# Effect of Chain Structure on the Various Properties of the Copolymers of Fluorinated Norbornenes with Cyclooctene

**DOI:** 10.3390/polym15092157

**Published:** 2023-04-30

**Authors:** Olga A. Adzhieva, Maria L. Gringolts, Yulia I. Denisova, Georgiy A. Shandryuk, Ekaterina A. Litmanovich, Roman Yu. Nikiforov, Nikolay A. Belov, Yaroslav V. Kudryavtsev

**Affiliations:** 1Topchiev Institute of Petrochemical Synthesis, Russian Academy of Sciences, Leninskii pr. 29, 119991 Moscow, Russia; adzhieva@ips.ac.ru (O.A.A.); denisova@ips.ac.ru (Y.I.D.); gosha@ips.ac.ru (G.A.S.); elitmanovich@yandex.ru (E.A.L.); nru@ips.ac.ru (R.Y.N.); belov@ips.ac.ru (N.A.B.); 2Faculty of Chemistry, Lomonosov Moscow State University, Leninskie Gory 1, Bld. 3, 119991 Moscow, Russia; 3Frumkin Institute of Physical Chemistry and Electrochemistry, Russian Academy of Sciences, Leninskii pr. 31, 119071 Moscow, Russia; 4ESPCI Paris, PSL Research University, 75005 Paris, France

**Keywords:** cross-metathesis, ring-opening metathesis polymerization, block copolymers, fluorinated polynorbornenes, chain structure, self-assembly, gas permeability

## Abstract

Fluorinated polymers are attractive due to their special thermal, surface, gas separation, and other properties. In this study, new diblock, multiblock, and random copolymers of cyclooctene with two fluorinated norbornenes, 5-perfluorobutyl-2-norbornene and N-pentafluorophenyl-exo-endo-norbornene-5,6-dicarboximide, are synthesized by ring-opening metathesis copolymerization and macromolecular cross-metathesis in the presence of the first- to third-generation Grubbs’ Ru-catalysts. Their thermal, surface, bulk, and solution characteristics are investigated and compared using differential scanning calorimetry, water contact angle measurements, gas permeation, and light scattering, respectively. It is demonstrated that they are correlated with the chain structure of the copolymers. The properties of multiblock copolymers are generally closer to those of diblock copolymers than of random ones, which can be explained by the presence of long blocks capable of self-organization. In particular, diblock and multiblock fluorine-imide-containing copolymers show a tendency to form micelles in chloroform solutions well below the overlap concentration. The results obtained may be of interest to a wide range of researchers involved in the design of functional copolymers.

## 1. Introduction

Fluorine-containing polymers display outstanding properties, such as thermal, chemical and oxidative resistances; a low refractive index, permittivity, and water absorption; excellent durability; and high gas permeability [1,2,3,4,5,6,7]. Nowadays, they are widely used as the components of coatings, fuel cell membranes, lithium–ion batteries, optical devices, organic electronics, etc. [8]. Many publications are devoted to the synthesis of fluorinated copolymers with a random and controlled chain structure, as discussed in reviews [9,10,11].

Of particular interest are block copolymers that exhibit spatially organized chemical specificity due to the tendency toward microphase separation and self-assembly into nanosized morphologies [12,13]. Among those, a special position belongs to multiblock copolymers that reveal the individual characteristics of their constituents and possess mechanical properties superior to those of homopolymer blends or di/triblock copolymers of a similar chemical composition. Since the synthesis and characterization of multiblock copolymers are often challenging and/or time-consuming, the detailed studies of their properties are not so numerous. Nevertheless, the research interest in this area steadily increases with the development of new synthetic approaches and techniques [14]. These include living or controlled anionic, cationic, radical, and olefin metathesis polymerizations, especially reversible addition–fragmentation chain transfer polymerization [15,16], copper-mediated radical polymerization [17,18,19], polycondensation [20], chain-shuttling polymerization [21], and inter-macromolecular reactions [22,23,24].

Fluorine-containing multiblock copolymers are being actively studied as ion-conducting media, such as proton exchange membranes [25,26]. The introduction of fluorine atoms considerably increases the contrast between hydrophilic and hydrophobic moieties, thus promoting a well-developed microphase separation [27,28,29]. Of interest are also fluoro-copolymers containing blocks with different degrees of flexibility that are prone to the liquid–crystalline type of self-assembly [30].

Recently, our studies have focused on the development of a new method for the synthesis of statistical multiblock copolymers. The approach is based on the interchain cross-metathesis reaction between the homopolymers containing C=C double bonds in their main chains, namely polynorbornenes, polyalkenamers, polydienes, and their derivatives [31,32,33,34,35]. The reaction takes place in the presence of well-defined Ru-carbene Grubbs’ catalysts. It seems interesting to expand the scope of this method to obtain fluorinated copolymers. As far as we know, there are practically no examples of fluorine-containing multiblock copolymer synthesis by the olefin metathesis reaction. In the 1990s, Feast et al. obtained diblock copolymers by the sequential polymerization of various fluorine-substituted norbornenes and norbornadienes [36,37]. In the presence of Schrock’s Mo-catalysts, the reaction proceeded in a living fashion and resulted in narrow dispersity copolymers. More recent examples of fluorine-containing block copolymer synthesis were also represented mainly by norbornene derivatives—in particular, N-substituted norbornene-5,6-dicarboximides [38,39,40,41,42].

The choice of norbornenes as monomers can be naturally explained. First, the strained norbornene molecule is one of the most active substrates in ring-opening metathesis polymerization (ROMP). This is especially important, because the electron-withdrawing fluorine atom reduces the activity of C=C double bonds in ROMP. Second, the side processes in ROMP are intra- and interchain transfers that, respectively, lead to cyclo-oligomer formation and reshuffle the growing copolymer blocks. For a polymer chain with bulky norbornene units, both side reactions are hindered, which facilitates the synthesis of precise block structures and provides a narrow dispersity in the copolymers. Accordingly, the block copolymer synthesis starts with a sterically more complex monomer, and both stages are carried out in the shortest possible time in the presence of an active catalyst. For instance, the synthesis of a block copolymer of fluorine-containing norbornene and cyclooctene was conducted using Schrock’s Mo-carbene complex and started with the polymerization of the substituted norbornene [43]. The deactivating effect of a fluorine atom on a C=C double bond can be reduced by increasing the distance between them, which was realized in N-substituted norbornene-5,6-dicarboximides [38,39,40,41,42] and exo-tricyclononenes [44,45,46,47]. An opposite effect was demonstrated in ref. [48], where the fluorine atoms located at double bonds prevented the homopolymerization of a norbornene derivative and enforced the formation of its alternating copolymer with dihydrofuran. The synthesized metathesis fluorine-containing block copolymers revealed microphase separation [49,50] and showed a trend toward self-assembled aggregation [51,52,53,54]. The latter effect was reported even for the random copolymers [41].

In this study, new copolymers of two fluorine-containing norbornenes and cyclooctene were synthesized by the macromolecular cross-metathesis, ring-opening metathesis copolymerization, and sequential addition of monomers in ROMP. Depending on the synthetic technique, multiblock, random, and diblock copolymers were, respectively, obtained. The influence of a chain structure on the polymer characteristics was studied with respect to the thermal, surface, gas permeation, and solution properties.

## 2. Materials and Methods

### 2.1. Materials

All manipulations involving air- and moisture-sensitive compounds were carried out in oven-dried glassware using dry solvents and standard Schlenk and vacuum-line techniques under an argon atmosphere. Cis,cis-1,5-cyclooctadiene (Aldrich, St. Louis, MO, USA) was dried over sodium and stored under argon. Grubbs’ catalysts of the 1st (G1), 2nd (G2), and 3rd (G3) generations (Aldrich) were used without purification. An inhibitor of oxidation, 2,2′-methylene-bis(6-tert-butyl-4-methylphenol) (Aldrich), and ethyl vinyl ether were used as supplied. Other reagents and solvents (Aldrich) were purified by common procedures.

### 2.2. Monomer Synthesis

5-(1,1,2,2,3,3,4,4,4-nonafluorobutyl)bicyclo[2.2.1]hept-2-ene (5-perfluorobutyl-2- norbornene) (NBF) and N-pentafluorophenyl-exo-endo-norbornene-5,6-dicarboximide (NBFD) monomers were synthesized by the procedures described in refs. [55,56], respectively. The synthesis of NBFD is exemplified below. At the first stage, a solution of pentafluoroaniline (7.58 g, 0.0414 mol) in dry methylene chloride (20 mL) was added dropwise to a stirred solution of norbornene-5,6-dicarboxylic anhydride (6.66 g, 0.0406 mol) in dry methylene chloride (50 mL). The resulting mixture was heated at 45 °C for 5.5 h, then concentrated on a rotor evaporator to obtain white amic acid crystals (*R_f_* = 0.83, ethyl acetate/hexane = 1/1 vol.). The amic acid yield was 13.4 g (99%). At the second stage, the acid obtained was dissolved in acetic anhydride (28.5 mL), and anhydrous sodium acetate (3.315 g) was added. The resulting suspension was heated for 25 h at 90–110 °C. After cooling to room temperature, the reaction mixture was washed with dilute HCl and extracted with diethyl ether. The ether extract was triple-washed with dilute HCl, saturated NaHCO_3_ solution, and distilled water; after which, the ether was evaporated to leave a grayish solid (10.84 g). The material was recrystallized twice from hexane to create white crystals (*T_m_* = 125.5 °C) with a yield of 8.13 g (62%).

### 2.3. Homopolymer Synthesis

Poly(1-octenylene) (polycyclooctene, PCOE), poly(5-perfluorobutyl-2-norbornene) (PNBF), and poly(N-pentafluorophenyl-norbornene-5,6-dicarboximide) (PNBFD) were synthesized by ROMP on Grubbs’ catalysts of the 1st and 3rd generations according to the procedures described in refs. [55,56,57], respectively. Their NMR spectra are shown in Appendix A Appendix A. The procedure for the synthesis of PNBFD at the molar ratio [NBFD]:[G1] = 180:1 is as follows. The G1 catalyst (2.7 mg, 0.0033 mmol) was placed into a two-necked flask with magnetic stirrer, and dry chloroform (0.20 mL) was added. A solution of NBFD (193 mg, 0.587 mmol) in dry chloroform (0.74 mL) prepared separately was added to the catalyst solution, and after 24 h, the reaction was stopped with ethyl vinyl ether. Upon the addition of chloroform and an oxidation inhibitor, the polymer product was precipitated into ethanol and dried under reduced pressure to a constant mass. The yield of PNBFD was 0.186 g (96%).

### 2.4. Synthesis of NBFD-COE and NBF–COE Copolymers with Different Chain Structure

*Random copolymers* were obtained by copolymerization of the initial monomers. The procedure is exemplified for the NBFD–COE copolymer obtained at the molar ratio [COE]:[NBFD]:[G1] ≈ 350:350:1. The solution of NBFD (151 mg, 0.468 mmol) in dry chloroform (0.27 mL) was prepared in a two-necked flask with a magnetic stirrer under an argon atmosphere, and cyclooctene (60 mg, 0.545 mmol) was added. Then, a separately prepared 0.10 mL of G1 (0.0014 mmol) solution in dry chloroform was added to the mixture of monomers. After 1 h, ethyl vinyl ether, chloroform, and an oxidation inhibitor were added to the reaction mixture. The NBFD–COE copolymer was precipitated into ethanol and dried under reduced pressure to a constant mass. The copolymer yield was 182 mg (87%). ^1^H and ^13^C NMR spectra of the random copolymers are shown in Appendix A.

*Multiblock copolymers* of the same monomers were obtained by cross-metathesis according to the procedure described in ref. [34]. Immediately before cross-metathesis, the initial homopolymers were purified from catalyst and inhibitor residues by reprecipitation of their solutions in chloroform (C_6_F_6_ for PNBF) into ethanol. The polymers were thoroughly dried under reduced pressure to a constant mass for at least 2 days. The following procedure describes the synthesis of the NBFD–COE copolymer obtained at the molar ratio [PCOE]:[PNBFD]:[G2] ≈ 600:200:1. The homopolymers PCOE (100 mg, 0.91 mmol) and PNBFD (100 mg, 0.30 mmol) were placed in a round-bottom two-necked flask and then dry chloroform (2.2 mL) was added under argon, and the system was left overnight. Then, it was degassed three times using the freeze–pump–thaw cycle technique before 0.12 mL of the G2 catalyst (0.0015 mmol) solution in dry chloroform was added. After a required time, the cross-metathesis was stopped by adding ethyl vinyl ether at a molar ratio of 500:1 to the catalyst and stirring the reaction mixture for 30 min. Then, the oxidation inhibitor was added, and after stirring for 0.5–1 h, the NBFD–COE copolymer was precipitated into ethanol and dried under reduced pressure to a constant mass. The NBFD–COE copolymer yield was 160 mg (79%). (For the NBF–COE copolymer: first, PNBF was dissolved in dry C_6_F_6_, and then, PCOE and dry chloroform were added one after another.) ^1^H and ^13^C NMR spectra of the multiblock copolymers are shown in Appendix A.

*Diblock copolymers* were synthesized as described in ref. [58]. The procedure is exemplified for the NBFD–COE copolymer obtained at the molar ratio [PCOE]:[PNBFD]:[G3] ≈ 1170:370:1. The solution of NBFD (170 mg, 0.52 mmol) in dry THF (0.86 mL) was prepared in a two-necked flask with a magnetic stirrer, and a 0.05 mL solution of the G3 catalyst (0.0014 mmol) in THF was added (the mixture turned orange). After 1 h, 0.20 mL of COE (180 mg, 1.64 mmol) was added, and after the next minute (the solution solidified and turned yellow), 4 mL of THF, ethyl vinyl ether, and an oxidation inhibitor were added. The NBFD–COE copolymer was diluted and precipitated into ethanol, then dried under reduced pressure to a constant mass. The yield was 287 mg (81%). ^1^H and ^13^C NMR spectra of the diblock copolymers are shown in Appendix A.

### 2.5. Characterization

Nuclear magnetic resonance (NMR) measurements were carried out at room temperature using a Bruker (Billerica, MA, USA) Avance III HD (400 MHz) spectrometer at 400.1 MHz (^1^H NMR) and 100.6 MHz (^13^C NMR) in CDCl_3_ solution. ^1^H and ^13^C chemical shifts in ppm relative to tetramethylsilane were determined using the signals of residual CHCl_3_ (7.28 ppm) and CDCl_3_ (77.23 ppm), respectively.

The molar mass distribution of the polymers was determined by gel permeation chromatography (GPC) on a modular high-pressure chromatograph equipped with a LabAlliance (Scientific Systems, Woburn, MA, USA) Series 1500 constant flow pump, refractometric LKB Bromma (Singapore) RI 2142 detector, and serially connected Waters (Milford, CT, USA) WAT054460 and Tosoh (Tosoh Bioscience, Tokyo, Japan) Biosep G3000HHR columns, with tetrahydrofuran (THF) as the solvent, a flow rate of 1 mL min^−1^, sample volume of 50 μL, and sample concentration of 1 mg L^−1^. The weight-average molar mass *M*_w_ and dispersity *Đ* were calculated by a standard procedure relative to polystyrene standards (Aldrich).

Differential scanning calorimetry (DSC) thermograms were recorded on the Mettler (Greifensee, Switzerland) DSC3+ at a rate of 10 °C min^−1^ under a nitrogen flow of 50 mL min^−1^ in a range from −100 °C to 100 °C. The data of the 2nd heating were processed using the STARe (Mettler Toledo, Columbus, OH, USA) service program. The accuracy of the measurements was 0.3 °C for temperature and 1 J g^−1^ for enthalpy.

Water contact angle measurements were performed using the sessile drop method. To this end, thin polymer films were prepared by dripping 0.5% chloroform solutions onto 1 × 1 cm^2^ silica wafers cleaned by a hot mixture of concentrated NH_4_OH and 30% H_2_O_2_ and deionized water and placed into a Laurell (Laurell Technologies, Lansdale, PA, USA) WS-650MZ-23NPP/OND spin coater. The substrate was rotated at a rate of 3000 rpm for 30 s. After drying, the film was placed underneath a syringe (*V* = 10 μL) with a 0.5 mm diameter needle, and a drop of water (4–5 μL) was dispensed onto the film surface. The contact angles were measured with an accuracy of 1° using a DigiMicro Mini USB microscope with a resolution of 2592 × 1944 pixels connected to a computer with S-EYE software (Shellfilm Technology, Shenzhen, China). The reported contact angle values were averages of at least three different areas of two samples for each polymer.

Films for gas penetration measurements were prepared by casting 3 wt% polymer solutions in THF onto a cellophane support, followed by the solvent evaporation. The detached films 30–40 μm thick were dried to a constant mass under residual pressure not exceeding 1 mm Hg. The gas permeability of the diblock D1 copolymer was measured on a free film and that of the random C3 and C3a copolymers on an ultrafiltration membrane made of polyacrylonitrile. The permeability and diffusion coefficients of H_2_, He, N_2_, O_2_, CO_2_, and CH_4_ gases in the polymer free-standing films were measured by the integral barometric method on a thermostated MKS Baratron (Andover, MA, USA) setup. The experiments were carried out at room temperature and pressure of 1–2 atm above the film and of ~10^–3^ mm Hg under it; therefore, the back diffusion of the penetrating gas was negligible. From the curve of gas leakage through the polymer film into the calibrated volume, the permeability coefficients *P* (by the slope of the linear dependence of the flow through the film upon reaching the stationary mass transfer mode) and diffusion coefficients *D* (by the Daynes–Barrer method, *D* = *l*^2^/(6θ), where *l* is the film thickness and θ is the so-called time lag) were determined. The experimental errors in measuring *P* and *D* were 5% and 10%, respectively. Accordingly, the error in calculating the solubility coefficients, *S* = *P*/*D*, was 20%.

Prior to light scattering measurements, the copolymer solutions in CHCl_3_ were filtered through Millipore filters with hydrophobic polytetrafluoroethylene membranes having a pore diameter of 0.22 μm. A Photocor Complex (Photocor Instruments, Moscow, Russia) laser light-scattering goniometer equipped with a HeNe laser (a wavelength of λ = 633 nm and an intensity of 10 mW) as a light source was used. In static (SLS) experiments, the weight-average molar mass *M*_w_ and the second virial coefficient A_2_ were found using the Debye equation *Kc*/*R*_θ_ = 1/*M*_w_ + 2A_2_*c*, where *K* = 4π^2^*n*^2^(*dn*/*dc*)^2^/(λ^4^*N*_A_) is the optical constant of the solution, *n* is the refraction index of a solvent (*n* = 1.4426 ± 0.0003 for CHCl_3_), *c* is the polymer concentration, and *N*_A_ is the Avogadro number. The Rayleigh ratio *R*_θ_ = α(*I*_s_ − *I*_solv_)sinθ allows for excluding the intensity of scattering by the solvent molecules, *I*_solv_, from the measured intensity of the solution scattering, *I*_s_, and α = 1.1416 × 10^−5^ (*n*_s_/*n*_tol_)/(*I*_t_(90°) − *I*_0_), where *I*_t_ is the signal of the light scattering by the standard sample (toluene) measured at the angle of 90°, *I*_0_ is the dark current signal, and *n*_s_ and *n*_tol_ are refraction indexes of a solvent and toluene, respectively. In dynamic (DLS) experiments, the time cross-correlation function *g*_2_ of the scattered-light intensity fluctuations was determined with a 288-channel Photocor-FC correlator board and treated with DynaLS software that applies the Pike eigenfunction truncation method with Tikhonov regularization to obtain hydrodynamic radius distributions.

## 3. Results and Discussion

### 3.1. Polymer Synthesis and Characterization

Figure 1 depicts monomers that were polymerized by ROMP into corresponding homopolymers in the presence of Ru-based Grubbs’ catalysts along the lines of refs. [55,56,57]. Among the monomers are cyclooctene (COE) and two fluorine-containing norbornenes, 5-perfluorobutyl-2-norbornene (NBF) and N-pentafluorophenyl -norbornene-5,6-dicarboximide (NBFD). The norbornenes were synthesized according to modified literature procedures [55,56], as described in Section 2.3. The conditions for the polymerization of COE, NBF, and NBFD are listed in Table 1, along with the structural characteristics of PCOE, PNBF, and PNBFD homopolymers. For the sake of further comparison with copolymers, PNBF/PCOE and PNBFD/PCOE 1:3 (mol) mixtures were prepared by coprecipitation.

It is seen that, under different ROMP conditions, homopolymers with high yields of 80–96% and various molar masses and *cis*/*trans* C=C ratios were obtained. Their molar masses predictably increase with the values of the monomer/catalyst ratio, since Grubbs’ catalysts are, in fact, initiators of the ROMP process. PCOE was synthesized on both G1 and G3 with good yields in 3 h. In the presence of more active G3, the content of the thermodynamically preferred *trans*-COE units in the polymer is much higher (87% vs. 60%). In the case of polynorbornenes, a polymer formed on G3 in just 1 h was further used for the diblock copolymer syntheses.

Copolymers of fluorine-containing norbornenes NBF and NBFD with COE were synthesized three ways, illustrated in Figure 2. Table 2 contains the information about the reaction conditions and structural characteristics of their products. All homopolymers and copolymers were characterized by ^1^H and ^13^C NMR (Appendix A), GPC (Appendix A), and DSC.

Random NBF–COE C1–C4 and NBFD–COE C5–C7 copolymers were obtained by adding the G1 catalyst to comonomer mixtures with various molar ratios of NBF(D) to COE. The molar ratio [NBF(D)]/[COE] = 1:3 corresponds to a 1:1 mass ratio, which is important for evaluating the copolymer properties. As seen from Table 2, the C1, C2, and C5–C7 copolymers synthesized in 1 h contained somewhat less NBF(D) than the initial comonomer mixtures, whereas C3 and C3a obtained under the same conditions in 2 h contained more of NBF. This indicates a lower activity of NBF(D) compared to COE in the copolymerization. As we will see in the next section, the thermal and crystalline properties of the copolymers depend significantly on the COE content in them.

Diblock NBF(D)–COE D1 and D2 copolymers were synthesized in the presence of the highly active G3 catalyst according to the procedure described in ref. [58]. ROMP of less-active NBF(D) monomers was carried out first; after 1 h, COE was added, and the reaction was stopped after 1 min to avoid the interchain cross-metathesis reaction to which the COE block is prone. The molar mass characteristics were *M*_w_ = 86 kg mol^−1^, *D* = 1.5 (Table 1) for the NBF block, *M*_w_ = 409 kg mol^−1^, *D* = 1.9 for the whole D1 copolymer, *M*_w_ = 169 kg mol^−1^, *D* = 1.9 (Table 1) for the NBFD block, and *M*_w_ = 527 kg mol^−1^, *D* = 2.7 for the whole D2 copolymer.

Multiblock NBF–COE M1–M3 copolymers were synthesized by the macromolecular cross-metathesis (MCM) of PNBF with PCOE in the presence of G1 (for M1) or G2 (for M2 and M3) catalysts. The reaction was carried out in the 1:1 (vol) mixture of C_6_F_6_ and CHCl_3_ solvents because of limited PNBF solubility in chloroform. As is known [33,59], at the early stage of MCM, the catalyst interacts with homopolymers by cutting their chains and forming Ru–carbene complexes at the ends of polymer chain fragments. This manifests itself by an apparent decrease in the viscosity of the reaction mixture, which was indeed observed in our experiments. It also explains a decrease in the molar mass of the M1–M3 copolymers relative to the parent PNBF and PCOE homopolymers (Table 1). The effect becomes more pronounced with increasing the amount of catalyst added. Note that the more active G2 catalyst is required in smaller amounts than G1.

Direct evidence of the copolymer formation is given by the heterodyad signals that appear in the NMR spectra of the reaction products. Unfortunately, the NMR spectra turned out to be uninformative both for the copolymerization of NBF with COE (Appendix A) and for the MCM of PNBF with PCOE (Appendix A). We were unable to detect new heterodyad signals in these spectra, possibly due to the overlap of homo- and heterodyad signals. As a result, the blocky structure of the different NBF–COE copolymers was analyzed only indirectly using the DSC and gas permeation data.

For the second polymer pair, PNBFD and PCOE, multiblock copolymer M4 was obtained by the MCM reaction in a chloroform solution in the presence of the G2 catalyst (Figure 2). In this case, ^13^C NMR spectra of the copolymers clearly contain the heterodyad signals (Figure 1, Appendix A). Their presence is most noticeable in the random C5 and C6 copolymers in the region of C=C bonds: 129.84, 129.49 (C^a^ in Appendix A), 133.30, 133.03, and 132.79 (C^b^ in Appendix A). For the C7 copolymer containing the significant excess of COE, the NBFD homodyad and heterodyad signals are very weak (Figure 1). Based on the integrals of the homo- and heterodyad signals, the average NBFD and COE block lengths, *L*_N_ and *L*_CO_, were calculated for the M4, C5, and C6 copolymers (the formula is presented next to Appendix A) and added to Table 2. Their values for C5 (*L*_N_ = 1.2, *L*_CO_ = 1.6) and C6 (*L*_N_ = 1.2, *L*_CO_ = 3.7) are even less than for the fully random copolymers of the corresponding compositions, which could mean that these copolymers have a tendency toward alternation. The NBFD and COE blocks in M4 are rather long (*L*_N_ = 39, *L*_CO_ = 92). It is worth noting that, at similar conditions, MCM between the equimolar amounts of unsubstituted PNB and PCOE on G2 resulted in a nearly fully random copolymer [31]; in the PNB–PCOAc–G2 system, a multiblock copolymer with the mean block length of 4 units was formed [34]; and even in the PNB(SiMe_3_)–PCOE–G1 system with a bulky Me_3_Si-substituent and less active catalyst, the resulting copolymers contained blocks of 7–21 units on the average, depending on the catalyst-to-polymer ratio [60]. Much larger block lengths in M4 indicate a significantly lower activity of PNBFD in MCM with PCOE than in the above cases, which can be related to the immiscibility of fluorinated norbornene with octenamer blocks, as well to the steric restrictions imposed by the bulky NBFD units.

Similar to the homopolymers, the multiblock copolymers obtained on G1 or G2 in the long (24 h) polymerization process demonstrate the dominance of *trans*-COE (about 80%) and *trans*-NBF(D) (about 70%) units.

### 3.2. Thermal and Surface Properties

The second heating DSC curves of the representative samples are collected in Figure 2 for the polymers based on the (a) NBF/COE and (b) NBFD/COE monomer pairs. Table 3 lists the numerical values of the glass transition (*T_g_*) and melting (*T_m_*) temperatures, the enthalpy of fusion (Δ*H*) and degree of crystallinity (DC) of the samples that reveal crystallinity, and the water contact angle (WCA) measured by the sessile drop method.

PCOE synthesized on G1 is a semicrystalline polymer with a low *T_g_* = −81 °C and a broad melting peak that spans from −40 °C to 40 °C with a maximum at 35 °C. Since 60% of its double bonds are in the *trans*-configuration (Table 1), only corresponding monomer units can crystallize [61] with the molar entropy of fusion −Δ*H*_μ_ = 20.16 kJ·mol^−1^ [62]. If G3 is used instead of G1 for ROMP, the resulting polymer contains more *trans*-COE units, which form in longer average sequences and thus crystallize into thicker crystallites. As a result, the melting temperature of PCOE increases up to 65 °C.

PNBF and PNBFD are amorphous polymers that undergo glass transition near 65 °C and 200 °C, respectively. Two glass transitions and one melting endotherm are visible on the DSC curves for both PNBF/PCOE (G1) and PNBFD/PCOE (G1) mixtures.

The NBF−COE (Figure 2a) and NBFD−COE (Figure 2b) copolymers inherit the properties of the parent polymers, but their characteristics depend on both the norbornene type and copolymer chain structure. The degree of crystallinity values listed in Table 3 are calculated per one *trans*-COE unit using the relation DC = (Δ*H/*Δ*H*_μ_)(μ/*m*)/ϕ*_tr_*_-COE_, where μ is the average molar mass of the monomer unit, *m* is the sample mass, and ϕ*_tr_*_-COE_ is the molar fraction of the *trans*-COE units in the (co)polymer. All of the random C1–C4 copolymers are semicrystalline, which corroborates the results of our earlier study of crystallinity in norbornene–cyclooctene copolymers [63], where we found that only isolated *trans*-COE units cannot be included in polymer crystallites. At the same time, their DC and *T_m_* are considerably lower than for PCOE, PNBF–PCOE, and other types of NBF−COE copolymers (except for C4, which contains ca. 95% of COE units), which stems from the combination of a relatively low content of *trans*-COE units and their statistical distribution along the chains. The glass transition temperature of the COE units for the C2 (−69 °C), C3 (−67 °C), and C3a (−75 °C) copolymers is still rather close to that of the COE units in PCOE (−81 °C), which implies that these copolymers are in a microphase-separated state. Only for C1 (−8 °C), it is close to the prediction of the Fox relation *T_g_* = *w*/*T_g_*_COE_ + (1 − *w*)/*T_g_*_NBF_, where *w* is the mass fraction of the COE units [64], which gives ca. −5 °C, thus indicating that the spatial distribution of the units in C1 is uniform.

Diblock D1 and multiblock M1–M3 NBF−COE copolymers possess elevated (near 50%) degrees of crystallinity provided by high (about 80%) fractions of *trans*-COE units in the corresponding blocks. As a result, the glass transition becomes less pronounced, and DSC detects the *T_g_* of the COE units in only one of these polymers (M1). Note that the *T_g_* of the NBF units can be masked by the COE melting endotherm. The melting temperature for D1 (60 °C) is close to that of PCOE synthesized on the same G3 catalyst. A decrease in the *T_m_* for the multiblock copolymers reflects the fact that the size of the *trans*-COE crystalline lamellae becomes more limited in the presence of NBF units in addition to *cis*-COE ones.

Qualitatively similar conclusions on the influence of the distribution of comonomer units on polymer crystallinity were previously made by Pitet et al. [58] for the copolymers of COE with phenylcyclooctene and by us [65] for the copolymer of cyclododecene with norbornene.

An influence of the bulky dicarboximide ring on the polymer thermal properties is evident. It results not only in the increase of the *T_g_* of the NBFD units by ca. 150 °C relative to the NBF ones (cf. Figure 2a,b) but also in a considerable drop of crystallinity in NBFD–COE copolymers. Random C5 and C6 copolymers are completely amorphous and characterized by a single *T_g_*, which is consistent with their short block structure reported by NMR (Table 2). Their glass transition temperatures can be approximated by the Fox relation that gives 63 °C and −5 °C instead of the measured 53 °C and 15 °C, respectively. The agreement is not very good in absolute terms, but note that the difference in the *T_g_* of the PCOE and PNBFD homopolymers is almost 300 °C.

On the contrary, diblock D2 and multiblock M4 NBFD–COE copolymers reveal two glass transition temperatures close to the *T_g_* values of the PNBFD and PCOE homopolymers. They retain crystallinity due to the high content of the *trans*-COE units, but the melting temperature and, hence, the mean crystallite size are noticeably lower than in the D1 and M1−M3 NBF−COE copolymers of the corresponding chain structure (Table 3). The degree of crystallinity remains rather high (47%) in D2 but drops to 30% in M4, though it has long COE blocks of 92 units on average (Table 2). Note, however, that the mean length of the crystallizable *trans*-COE sequences is much shorter.

Despite the common belief that fluorine atoms bring hydrophobicity, PNBF with a water contact angle (WCA) of 107 °C is only a little more hydrophobic than PCOE (97 °C), whereas PNBFD even reveals hydrophilicity (82 °C). In the mixtures, the WCA value is determined by a more hydrophobic component (106 °C for PNBF–PCOE and 96 °C for PNBF–PCOE) that segregates into the air–polymer interface in the course of the film preparation. The WCA for the copolymers predictably falls into the interval between the corresponding homopolymers. The polymers containing NBFD and/or COE units can be arranged by decreasing their WCA values, as PCOE (G1) > PNBFD/PCOE > D2 > M4 > C6 > C5 > PNBFD (G1) > PNBFD (G3). Shortening the block lengths suppresses their surface segregation, so that the random C5 and C6 copolymers are characterized by the WCA values of 91 °C and 88 °C, which are close to the arithmetic mean of the contact angles for its components.

### 3.3. Gas Transport Properties

The gas permeability of the PNBF homopolymer (*P*_CO2_ = 54 Barrer) [55] is known to be higher than that of PNBFD (*P*_CO2_ = 35 Barrer) [66]. Therefore, we chose NBF–COE copolymers—in particular, the random C3 and C3a and diblock D1 copolymers—to study the relation between their chain structure and gas permeability properties. Multiblock M1–M3 copolymers were not suitable for the preparation of free-standing films, presumably because of their insufficiently high molecular masses. Since, at room temperature, both C3 and C3a are in a rubbery, melted state, their gas permeability was measured on a polyacrylonitrile ultrafiltration membrane, whereas, in the case of D1 with *T_m_* = 60 °C, a free-standing film was prepared. The results for PNBF, C3, C3a, and D1 are summarized in Table 4, along with the literature data for polyethylene (PE), polybutadiene (PB1 and PB2), polypentenamer (PPM), and polydimethylsiloxane (PDMS).

The gas permeability coefficients of glassy amorphous PNBF are considerably lower than those of rubbery amorphous PDMS, PPM, and PB1; among which, the permeability drops with an increase in the glass transition temperature. However, the random C3 and C3a copolymers have a similar permeability as PB1, since they contain an excess of COE units, a *T_g_* of which is close to that of PB1, and which are already amorphous at room temperature. These copolymers demonstrate a significant increase in the permeability coefficients for the series of hydrocarbons that is similar to PPM behavior.

The gas permeability decreases for semicrystalline polymers, and the results for the diblock D1 copolymer are comparable to the literature data for PE and (*trans*- 90%) PB2. In that case, the permeability coefficients can be recalculated to the amorphous part of a polymer, assuming that its crystalline part is impermeable, as was argued in refs. [67,68] for PE and PB2. Upon this procedure, the data for D1 become rather similar to those for PNBF, because the amorphous part of the diblock copolymer consists, to a large extent, of NBF units.

Thus, we can conclude that the diffusion of gases in the random C3 and C3a copolymers occurs through a melted matrix consisting of NBF and COE units, whereas, in the case of the diblock copolymer D1, the penetrants mainly diffuse through glassy NBF domains. This is supported by Table 5, where the measured values of the gas diffusion coefficients *D_eff_*· are presented; the D1 copolymer looks similar to PNBF and PE, whereas the C3 and C3a copolymers are ahead of not only all semicrystalline polymers but also of fully amorphous PB1, second only to highly permeable PDMS.

It was also interesting to study the separation properties of polymers containing fluorine in a repeating norbornene-based unit. The data on the ideal selectivities, α*_P_*, for three common gas pairs are presented in Table 6 for PNBF, C3, C3a, D1, and two previously studied polymers, PE and PPM. One can see that nearly the same value of α*_P_* ≈ 2.9 for the O_2_/N_2_ pair is observed in all norbornene-based polymers, as well as in PE and PPM. However, fluorine does have influence on the ideal selectivities for the He/H_2_ and N_2_/CH_4_ pairs. In the PNBF homopolymer (31 mol% of F), they are equal to 1.6 and 1.0, respectively, which corresponds to α*_P_* values typical of highly fluorinated and perfluorinated polymers [5]. In the C3, C3a, and D1 copolymers that contain three times less fluorine atoms, the ideal selectivities are nearly twice as less, yet higher than in hydrocarbon-based PE and PPM polymers. At the same time, there is not much difference between the selectivity values in the copolymers with different chain structures.

To summarize, at room temperature, the diblock NBF–COE copolymer contains a glassy and a semicrystalline block and, therefore, reveals gas permeation properties similar to PNBF, whereas the amorphous and rubbery random NBF–COE copolymers behave similar to amorphous PB and PPM. The fluorination of norbornene units leads to a higher ideal selectivity for the He/H_2_ and N_2_/CH_4_ gas pairs, depending on the fluorine content, regardless of its distribution along the polymer chains.

### 3.4. Light Scattering

Since PNBF is poorly soluble in chloroform, studies of the solution properties by static (SLS) and dynamic light scattering (DLS) were performed for the polymers based on the second monomer pair, NBFD and COE. Note that, according to the WCA measurements, PNBFD is weakly hydrophilic (82°) and PCOE is weakly hydrophobic (107°). We chose PNBFD synthesized on G1 at a polymer-to-catalyst molar ratio of 180:1; PCOE (G1); and three NBFD–COE copolymers: random C6 with very short blocks, multiblock M4 with long blocks, and diblock D2 (see Table 2 and Table 3 for their characteristics).

Table 7 contains the values of the refraction index increment, *dn*/*dc*, measured directly and of the weight-average molar mass, *M_w_*, and the second virial coefficient, *A*_2_, determined using the Debye equation (see Section 2.5) for the above homopolymers and copolymers. One can see that the molar mass values obtained by light scattering and GPC (Table 2) coincide only for the M4 copolymer (189 kg·mol^−1^ and 190 kg·mol^−1^). Noticeable deviations in the other cases can reflect the difference between the studied polymers and polystyrene standards used for GPC calibration. The positive values of *A*_2_ indicate that chloroform is a good solvent for all polymers, especially for the PCOE and C6 copolymer, yet not as good for PNBFD and the M4 and D2 copolymers, which contain long sequences of NBFD units.

Dynamic light scattering from dilute (1.3–2.6 g·L^−1^, blue curves in Figure 3) solutions expectedly yields a monomodal distribution over the hydrodynamic radius, *R_h_*, with a mean value between 7 and 21 nm that corresponds to isolated polymer coils. Increasing the polymer concentration up to 3.7–5.2 g·L^−1^ (red curves) does not change the picture qualitatively for both homopolymers and the random C6 copolymer, but it leads to a bimodal distribution for the diblock D2 and multiblock M4 copolymers. In the latter case, this becomes more evident when the polymer concentration increases up to 7.4 g·L^−1^ (green curve). Apart from individual macromolecules, the solutions of D2 and M4 contain micelles with *R_h_* of 100–200 nm, as schematically shown in Figure 3. Note that all these solutions are well below the overlap concentration *c** = 3*M_w_*/(4π*N*_A_*R_g_*^3^), which can be estimated for D2 (the polymer with the highest *M_w_* = 385 kg·mol^−1^, *R_g_* = 0.7 *(R_h_*)*_av_* = 16.8 nm, and *N*_A_ = 6 × 10^23^·mol^−1^) to be 32 g·L^−1^. It is also worth mentioning that a considerable growth in the *R_h_* value for PCOE was found with DLS only when the polymer concentration attained 30 g·L^−1^ [59].

The observed partial aggregation of the D2 and M4 copolymers is very moderate. Indeed, the amplitudes of the primary and secondary peaks in Figure 3 are comparable, whereas the Rayleigh scattering intensity from a spherical particle increases proportionally to the sixth power of its radius. Thus, most of the copolymer chains still exist as separate coils in the studied concentration range, yet the tendency toward micellization should increase as the polymer concentration approaches *c**. Since it reveals itself in the copolymers with long blocks, the formation of micelles can be due to the immiscibility of NBFD and COE blocks, which is corroborated by two glass transitions detected in the D2 and M4 (not in C6) copolymers (Table 3).

Since CHCl_3_ is a better solvent for PCOE than for PNBFD, such micelles should consist of a NBFD core and COE shell. Micellization diminishes the amount of thermodynamically disadvantageous contacts between NBFD and COE units by localizing them at the interface between the corresponding domains. Since the blocks in a diblock chain are generally longer than in a multiblock one, they should separate easier, which is confirmed by the observation that the micelles in the D2 copolymer appear at a lower concentration than in M4 (Figure 3).

As shown in refs. [71,72], additional information on the self-assembling of copolymers in solutions can be obtained from computer simulations. We can also assume that D2 and M4 copolymers will undergo microphase separation in the bulk.

## 4. Conclusions

In this paper, Grubbs’ catalysts of three generations were used to synthesize polymers based on cyclooctene and one of two fluorinated norbornenes. Homopolymers, random copolymers, diblock copolymers, and, for the first time, statistical multiblock copolymers were obtained and their chain structure (by GPC and NMR), thermal (by DSC), surface (by sessile drop method), bulk (by measuring gas permeability and diffusivity), and solution (by SLS and DLS) properties were compared. We found that the resulting copolymer characteristics depend on the distribution of comonomers with very different chemical natures along the macromolecules. The diblock copolymers combine the crystallinity of polycyclooctene with the glassy amorphous characteristics of polynorbornenes, and the properties of the parent homopolymers are not influenced much by their connectivity. On the contrary, the random copolymers with short blocks possess averaged characteristics. As a result, the coexistence of crystalline and glassy domains in diblock copolymers can be replaced with an amorphous rubbery state in random ones at the same temperature. Multiblock copolymers take an intermediate position, being, in general, similar to diblocks but with a weakened tendency to block segregation. The attainability of particular characteristics depends on the possibility to control at least the composition and mean block lengths during the course of the copolymer synthesis.

## Data Availability

The data presented in this study are available on request from the corresponding authors.

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
