# Peer review of "Effect of Chain Structure on the Various Properties of the Copolymers of Fluorinated Norbornenes with Cyclooctene"

_polymers, 2023, doi:10.3390/polym15092157_

Round 1
Reviewer 1 Report
The correlation between the chain structure of polymers, self-assembly and thermal properties, surface properties, gas permeation, and solution properties of new materials was addressed.
The topic is original, and includes new and useful insights both for synthetic chemists as well as new information for theoretical chemists.
The research brings new fundamental scientific insights, through the synthesis and full characterization of various fluorine-containing polymers.
In order to better address the self-organization of multiblock copolymers and diblock copolymers, some structure simulation/Molecular modelling into supramolecular polymers would be useful to perform (see DOI: 1186/1752-153X-6-91; DOI: 10.1016/j.progpolymsci.2017.04.003). One graph representing the self-assembly is useful and should be inserted.
The results of the molecular modeling should be correlated with the properties of the compounds.
For the structure simulation/Molecular modelling, new references should be included.
The article is well written, summarizes a lot of valuable information in its figures and tables, and contains a lot of work.
Author Response
We are grateful to the Reviewer 1 for his valuable remarks. We agree with the Reviewer that computer simulations can shed more light on the solution behavior of our copolymers. In this regard, we have added a corresponding paragraph to the end of the Results and Discussion section, with two new references to the papers [71,72] on the modeling of copolymer self-assembly in solutions. Also we depicted the supposed structure of these solutions directly in the DLS plots in Figure 3 and expanded the caption to this Figure. Note that any simulation within the current paper is hardly possible because it requires thorough preliminary parametrization of the model to give reliable results.
Reviewer 2 Report
In the submitted manuscript by Kudryavtsev and co-workers, copolymers of cyclooctene (COE) with two fluorinated norbornene monomers, 5-perfluorobutyl-2-norbornene (NBF) and N-pentafluorophenyl-exo-endo-norbornene-5,6-dicarboximide (NBFD), are prepared using Grubbs catalysts. Three different types of NBF-COE or NBFD-COE copolymers – random copolymer, diblock copolymer and multiblock copolymer – are synthesized by ring-opening metathesis polymerization (ROMP), sequential monomer addition of ROMP, and macromolecular cross-metathesis, respectively. Notably, fluorine-containing multiblock copolymers are synthesized by olefin metathesis for the first time. The properties of obtained copolymers as well as corresponding homopolymers are thoroughly characterized and compared by a variety of analytical techniques, including chain structure (by GPС and NMR), thermal (by DSC), surface (by water contact angle), bulk (by gas permeability and diffusivity), and solution (by SLS and DLS) properties. These properties are related to the chain structures of the copolymers. In general, the multiblock copolymers exhibit properties that are intermediate between those of the diblock and random copolymers, but are closer to those of the diblock ones. Overall, the authors have synthesized different types of fluorine-containing copolymers, and thoroughly studied the relationship between their properties and the chain structures via various characterization techniques. The manuscript is well organized and easily understood, with rigorous experimental designs, clear arguments and well-supported conclusions. This work could be suitable for Polymers, after a few minor issues are addressed.
· Line 112 – “heptane-2” should be “hept-2-ene”
· Table 2 – for polymer M1-M4, are the [mon]/[cat] ratios and feed ratios based on the amount of repeating units in the macromolecules used in the reactions? This information should be clarified.
· Line 402 – Does “dicarboximide cycle” mean “dicarboximide ring”?
· Reference 35 – Page number and DOI is missing.
· Reference 38 – Journal name is missing.
Author Response
We are grateful to the Reviewer 2 who has highly rated our work. We introduced all of his suggestions, in particular:
(1) Replaced “heptene-2” with “hept-2-ene” on page 3; (2) Extended the footnote to Table 2 to clarify that the [mon]/[cat] ratios and feed ratios are based on the amount of double bonds in the substances used in the reactions; (3) Replaced “dicarboximide cycle” with “dicarboximide ring” on page 11; (4) Corrected the refs. 35 and 38 in the reference list.